# Multiple Mitochondrial Dysfunction Syndrome Type 3: A Likely Pathogenic Homozygous Variant Affecting a Patient of Cuban Descent and Literature Review

**DOI:** 10.3390/genes13112044

**Published:** 2022-11-06

**Authors:** Steven H. Lang, Francesca Camponeschi, Evan de Joya, Paulo Borjas-Mendoza, Mustafa Tekin, Willa Thorson

**Affiliations:** 1Dr. John T. Macdonald Foundation, Department of Human Genetics, Miller School of Medicine, University of Miami, Miami, FL 33136, USA; 2Magnetic Resonance Center, University of Florence, 50019 Florence, Italy

**Keywords:** multiple mitochondrial dysfunction syndrome, MMDS, IBA57, iron-sulfur clusters, leukoencephalopathy

## Abstract

Multiple mitochondrial dysfunction syndrome type 3 (MMDS3) is a rare mitochondrial leukoencephalopathy caused by biallelic pathogenic variants in *IBA57*. Here, we describe a homozygous variant in *IBA57*, (NM_001010867.2): c.310G>T (p.Gly104Cys), in a 2-month-old infant of Cuban descent who presented with a one-month history of progressive hypotonia, weakness, and episodes of upgaze deviation. This is the first report of a patient homozygous for this variant and the first report of MMDS3 in a patient of Hispanic descent described to our knowledge. Using in silico tools, we found that the variant resides in a putative mutational hotspot located in the neighborhood of a key active ligand required for iron-sulfur cluster coordination. In addition, while previous case reports/series have reported the variable phenotypic features of the disease, the incidence of these features across the literature has not been well described. In order to construct a clearer global picture of the typical presentation of MMDS3, we reviewed 52 cases across the literature with respect to their clinical, biochemical, genotypic, and neuroradiographic features.

## 1. Introduction

The multiple mitochondrial dysfunction syndromes 1–6 (OMIM#: 605711, 614299, 615330, 616370, 617613, 617954) are an exceedingly rare group of mitochondrial leukoencephalopathies caused by pathogenic variants in the *NFU1*, *BOLA3*, *IBA57*, *ISCA2*, *ISCA1,* and *PMPCB* genes, respectively [1,2,3,4,5]. These genes encode the critical machinery required for the final stages of mitochondrial iron-sulfur (Fe-S) cluster assembly and maturation [6,7]. Fe-S clusters are evolutionarily ancient prosthetic groups required for catalysis and electron transfer in ubiquitous metabolic processes, including the Krebs cycle and the mitochondrial respiratory chain. Functional IBA57 is necessary for the maturation of cubane [4Fe-4S] clusters found in complexes I and II of the respiratory chain, mitochondrial aconitase, as well as lipoic acid synthase, which is required for lipoylation of pyruvate dehydrogenase and alpha ketoglutarate dehydrogenase [8,9]. Biallelic pathogenic variants in the *IBA57* gene have been shown to cause a spectrum of diseases, including multiple mitochondrial dysfunction syndrome type 3 (MMDS3) in 52 patients [1,10,11,12,13,14,15,16,17,18,19,20,21] as well as a SPOAN (OMIM# 609541)-like phenotype in 11 members of a single family [22]. MMDS3 most commonly manifests in late infancy with regression of psychomotor milestones and white matter signal abnormalities on MRI, often with progression to a cavitating leukoencephalopathy, a radiographic hallmark of mitochondrial encephalopathies [23,24]. The clinical course is variable even among affected siblings, with some patients achieving a stable disease course and others experiencing progressive decline, often punctuated by episodes of febrile illness and ultimately demise in infancy. Here, we report a homozygous likely pathogenic variant in *IBA57* (NM_001010867.2): c.310G>T (p.Gly104Cys) in a 2-month-old infant of Cuban descent who presented with one month of progressive hypotonia, weakness, and episodes of her eyes “rolling into the back of her head”. While this variant has been previously reported by Sato et al. (2021) in a Japanese patient with a compound heterozygous genotype, this is the first report of a patient homozygous for this variant and the first report of MMDS3 in a patient of Hispanic descent described to our knowledge. Furthermore, we use in silico tools to assess the variant in the context of known pathogenic/benign variants in *IBA57*, as well as to propose plausible mechanisms of pathogenicity. In addition, we present a review of the literature to summarize the clinical, biochemical, neuroradiographic, and genotypic findings of 52 patients with MMDS3. 

## 2. Materials and Methods

### 2.1. Mitochondrial Genome Sequencing 

Genomic DNA was obtained from the proband’s buccal swab, and the entire mitochondrial genome was amplified and sequenced using next-generation sequencing (NGS) technology. The DNA sequence was assembled and analyzed with reference to the revised Cambridge Reference Sequence (NC_012920) as well as reported variants in the MITOMAP database (https://www.mitomap.org/MITOMAP, accessed on 1 June 2022). This method is able to detect mtDNA point variants at >1.5% heteroplasmy and large mtDNA deletions at >5% heteroplasmy. 

### 2.2. Leukodystrophy and Leukoencephalopathy Panel

Genomic DNA obtained from the proband’s blood sample was enriched for targeted regions of well-characterized genes involved in leukodystrophies/leukoencephalopathies. The panel utilizes Illumina technology and provides ~98.3% coverage of all coding exons of selected genes plus 10 bases of flanking non-coding DNA. Other non-coding regions were included in select genes in which non-coding variants have been implicated in disease. Sequences were aligned to the hg19 genome build, and variants were reported as per human genome variation society guidelines (HGVS) (https://www.hgvs.org/, accessed on 1 July 2022). 

### 2.3. Whole Exome Sequencing Trio Analysis 

Genomic DNA was obtained from the proband’s and parent’s buccal swabs. The proband’s and parental DNA were enriched for coding regions and splice junctions. Targets were sequenced using Illumina technology, and sequences were aligned to reference sequences based on NCBI RefSeq transcripts and the hg19 genome build. Variants were filtered using proprietary software, and sequence variants were reported in accordance with HGVS guidelines. Reported clinically significant variants were confirmed by Sanger sequencing in the proband as well as both parents for the purpose of allowing better interpretation of the results from the affected individual.

### 2.4. Literature Review

We examined clinical reports of patients with *IBA57* variants causing MMDS3. We queried PubMed for the keyword “*IBA57*”, which returned 24 results from 2008 to 2022. We excluded 9 papers that lacked clinical data, 1 that was itself a review, and 1 that described childhood onset of a SPOAN-like phenotype. Patients with the SPOAN-like phenotype, associated with a splice-site variant in *IBA57*, were excluded as we feel it likely represents a distinct clinical entity due to its highly attenuated phenotype and later age of onset. A total of 13 papers describing a total of 51 cases affected by biallelic missense, nonsense, and frameshift variants were ultimately included in our review in addition to our original case report. 

With respect to the reporting of clinical findings, we compiled a list of clinical characteristics commonly described across case reports. Patients were coded as possessing the trait if explicit mention was made in the clinical report, patients were coded as not possessing the trait only if the explicit mention of trait absence was made in the clinical report, and were otherwise reported as missing data, resulting in different “N”s across different characteristics. 

With respect to the reporting of neuroradiographic findings, when presented with a detailed radiographic report, we considered the omission of neuroanatomical regions to mean the absence of involvement in that region. Conversely, when presented with terse descriptions of findings, we extracted pertinent positives and considered the omission of findings as missing data.

### 2.5. In Silico Analysis 

We used the UCSC genome browser tool (https://genome.ucsc.edu/, accessed on 1 August 2022) comparative genomics conservation track to obtain sequences of *IBA57* homologs in the region of the variant across species. 

The crystal structure of human IBA57 has been previously elucidated and is available on the RCSB protein databank (https://www.rcsb.org/structure/6QE3, accessed on 1 August 2022) [25,26,27]. We used PyMOL to visualize the impact of the variant on the IBA57 protein structure. 

In order to explore our variant in the context of the known mutational landscape of *IBA57*, we pulled variants from the case reports/series included in our review as well as pathogenic/likely pathogenic variants reported in the ClinVar database (if not already described in a published case report/series), as well as benign missense variants reported in the ClinVar database. 

We used the scaled point system described by Tagtivian et al. (2020) to classify the variant in accordance with ACMG guidelines [28]. 

## 3. Results

### 3.1. Clinical Report

A two-month-old female presented with a one-month history of progressive hypotonia, peripheral and central weakness, fatigue, loss of object tracking, and intermittent episodes of upward gaze deviation. The patient was born full term to a consanguineous 27-year-old G1P0 mother and a 50-year-old father of Cuban origin. At birth, the patient weighed 2.96 kg (27th percentile), was 49.6 cm long (60th percentile), and had a reported normal head circumference (measurement not available). Prenatal history and perinatal period were unremarkable. Prior to the onset of symptoms, the patient achieved milestones appropriate for a one-month-old and was generally healthy, with the exception of mild reflux treated with famotidine. No preceding febrile illness or vaccinations were noted prior to the onset of symptoms. On exam, weight, length, and head circumference were 4.60 kg (12th percentile), 58 cm (44th percentile), and 37.7 cm (17th percentile), respectively. An initial neurological exam revealed an overall paucity of movement, axial and appendicular hypotonia, poor suck reflex, and dysphagia. During the course of her admission, the patient also developed brief episodes of opisthotonos after crying. An ophthalmological exam revealed episodes of intermittent upgaze deviation lasting 1–2 s and no visual threat response. No additional eye movement abnormalities were observed, including nystagmus. No grossly dysmorphic features were present. Seizures were not reported or observed over the course of her admission. Biochemical studies were notable for two episodes of transient lactic acidosis, which correlated with periods of acute decompensation (3.6 mmol/L peak and 3.9 mmol/L peak (reference range: 0.7–2.1 mmol/L)). Acylcarnitine levels were within normal limits. MRI revealed widespread signal abnormalities involving subcortical white matter, periventricular white matter, and optic nerves, as well as a cystic signal abnormality within the medulla. Brain MR spectroscopy revealed a significant lactate peak. EEG was non-remarkable. Mitochondrial genome sequencing did not reveal any pathogenic variants or variants of unknown significance. The Prevention Genetics Leukodystophy/Leukoencephalopathy panel revealed a homozygous variant in *IBA57* (NM_001010867.2): c.310G>T (p.Gly104Cys). Trio exome analysis later revealed that these variants were in trans. Treatment was mainly supportive. The infant showed worsening hypotonia and impending respiratory failure requiring intubation and ventilatory support by 3 months old due to neuromuscular degeneration and ultimately expired after parents agreed to withdraw supportive care. 

### 3.2. In Silico Analysis

Structural modeling of the variant in PyMOL revealed that the substitution p.Gly104Cys is located in a flexible loop region within 11 angströms of the Cys259 residue, which is involved in the binding of a [2Fe-2S] cluster in wild-type IBA57 (Figure 1). The position is conserved across all species with known homologous sequences (Figure 2). 

With respect to our analysis of the variant in the context of the known mutational landscape of *IBA57* (Figure 3), we found that this variant was previously described by Sato et al. (2021) in a neuroradiology case report (although not reported in any database) describing a Japanese infant presenting with fetal growth restriction, apnea, lactic acidosis, and a subependymal pseudocyst with a fluctuating membrane on neuroimaging, who ultimately expired at 30 days of life. The patient had a compound heterozygous genotype (c.49_67dup(p.Leu23fs), c.310G>T(p.Gly104Cys)), and analysis of mitochondrial enzymes revealed extremely low activity of respiratory chain complexes I and II. 

Interestingly, we found that our variant appears to occur within a putative mutational hotspot of the IBA57 protein. In an eight amino acid residue range flanking our variant (residues 96–112), we identified seven known pathogenic variants (two frameshift, two nonsense, and three missense), three variants of uncertain significance (all missense), and no benign variants.

Included in this hotspot is a variant one amino acid residue downstream of our variant c.313C>T(p.R105W). The affected patient was a compound heterozygote (c.87ins_GCCCAAGGTGC(p.R30Afs*46); c.313C>T(p.R105W)) who presented with psychomotor regression at 8 months of age and eventually developed respiratory failure at 16 years of age. Analysis of mitochondrial enzymes revealed decreased activity of respiratory chain complexes I and II, as well as decreased levels of the lipoylated enzymes pyruvate dehydrogenase and alpha ketoglutarate dehydrogenase [20].

With respect to the ACMG classification of this variant, the (NM_001010867.2): c.310G>T (p.Gly104Cys) variant obtains a score of 8 (2 supporting plus 3 moderate); thus, re-classifying it from a variant of unknown significance to likely pathogenic (Table 1). This conclusion is also consistent with ACMG guidelines for likely pathogenic variants (≥3 Moderate (PM1–PM6) or 2 Moderate (PM1–PM6) and ≥2 supporting (PP1–PP5)). Lastly, the variant does not meet the criteria to support benign impact.

### 3.3. Literature Review

#### 3.3.1. Clinical Characteristics

We found that, where explicitly reported, 67.9% (19/28) of patients had normal development prior to symptom onset, with a median age of onset of 9 months old and a range of onset from birth to 22 months old (Table 2). Disease was more commonly reported in females at 54.9% (28/51) compared to males at 45.1% (23/51). With respect to ethnic background, disease was most commonly reported in East Asians, 74% (37/50). Our case is the first Hispanic patient described in the literature to our knowledge (Figure 4). Psychomotor regression was overwhelmingly the most reported presenting sign present in 81.8% (27/33) of cases, followed by decreased energy (12.1%), hypotonia at birth (9.1%), and visual impairment (6.1%). With respect to neurologic features, impaired motor function was a nearly universal finding in all but one patient who presented with isolated visual impairment and stable disease across his lifetime [15,29].

Hypotonia was present in 91.7% (11/12) of patients when explicitly mentioned and was the presenting sign at birth for three patients [1,13]. Intellectual disability was present in 80% (12/15) of patients. Epilepsy/seizures were reported in 35% (7/20) of patients. Visual impairment was noted in 46.7% (7/15) of patients. Apneic episodes and respiratory pump failure requiring ventilatory support were reported in six patients and were the cause of death in all cases where a proximate cause was explicated stated (3/3). Death was reported in 10 patients with a median age of 7 months old (range 6 days old to 27 months old). A gestalt of overall clinical progression was described in 22 cases, with 27.3% of patients reported as stable/improving, 22.7% had paroxysms of acute worsening followed by a period of stability, and 50% had continuous progressive worsening. The median time from diagnosis to the last follow-up was 33 months (*n* = 26), with a range of 1–328 months. Initial presentation and paroxysms of acute worsening were often preceded by acute febrile illness and, in two cases, vaccination.

#### 3.3.2. Neuroradiographic Findings

Periventricular white matter signal abnormalities were the most frequently reported findings present in 83.3% (20/24) of patients (Table 3), followed by signal abnormalities in parieto-occipital white matter in 66.7% (14/21) of patients, and the corpus callosum in 62.5% (15/24) of patients. White matter cavitations/cystic lesions were present in 81.5% (22/27) of reported cases. MRS was reported in nine cases and revealed a significant lactate peak in all nine cases.

#### 3.3.3. Biochemical Findings

Elevated CSF lactate was reported in 72.7% (8/11) of cases (Table 2). At least one measurement of serum lactate elevation was reported in 53.8% (7/13) of cases, although this was explicitly described as a transient elevation in two cases, including our own. When assayed from tissue samples, severely decreased activity of respiratory chain complexes I and II were found in 6/7 patients. Decreased activity of complex IV was found in 2/7 patients. Decreased activity of complex III was not detected in any patients. Elevated serum glycine levels were reported in 5/10 patients in whom serum amino acid levels were reported, and CSF glycine was elevated in 3/4 patients.

#### 3.3.4. Treatments

Treatment aimed at modifying disease progression was explicitly mentioned in 10 cases and consisted of unspecified mitochondrial and vitamin cocktails, riboflavin alone, riboflavin and coenzymeQ10, as well as vitamin cocktail in combination with injected and oral steroids. Other supportive care measures included tracheostomy for ventilatory support in the setting of respiratory pump failure, baclofen for the treatment of spasticity, as well as levetiracetam and carbamazepine for seizure control.

## 4. Discussion

In this report, we describe a homozygous variant in *IBA57*, *IBA57*: c.310G>T (p.Gly104Cys), causing MMDS3 in a two-month-old female. We found that this variant was previously described by Sato et al. (2021) in a patient with a compound heterozygous genotype (c.49_67dup(p.Leu23fs); c.310G>T(p.Gly104Cys)) and extremely low activity of respiratory chain complexes I and II. Admittedly, given the patient’s compound heterozygous genotype with a frameshift variant, it is not possible to know the precise contribution of the c.310G>T(p.Gly104Cys) variant. Our comprehensive review of all reported protein-coding variants in *IBA57* revealed a high density of pathogenic variants and an absence of benign variants in the neighborhood of the c.310G>T (p.Gly104Cys) variant, which may constitute a mutational hotspot. These findings, taken together with the patient’s highly specific phenotype and multiple lines of computational evidence supporting a deleterious effect, permit the promotion of this variant from a variant of unknown significance to likely pathogenic by ACMG criteria (score of 8).

Interestingly, the p.Gly104Cys substitution is located within 11 angströms of the Cys259 residue, which has been previously identified as the critical ligand required for [2Fe-2S] cluster coordination by IBA57 in the context of [2Fe-2S] ISCA2-IBA57 heterocomplex formation [25,27]. Variants in the neighborhood of this ligand have been previously described, including the p.Arg146Trp variant, which has been experimentally shown to inhibit heterocomplex formation [27]. We, therefore, propose two plausible mechanisms of action for this variant. First, the Gly104Cys variant may affect the interaction with ISCA2 or the cluster coordination properties of the ISCA2-IBA57 heterocomplex, either by forming disulfide bonds with the Cys259 residue of IBA57, which is located on a flexible loop region in the close proximity of the Gly104Cys variant, or with the conserved cysteines Cys79, Cys144, and Cys146 residues of ISCA2, or by directly interacting with the [2Fe-2S] cluster, acting as an alternative ligand, and promoting the formation of an aberrant [2Fe-2S] ISCA2-IBA57 complex structurally and functionally different from that formed by the wild-type protein. Second, the p.Gly104Cys variant could also induce changes in the 3D structure of IBA57 due to steric effects arising from the bulkier side chain of Cys with respect to Gly. This potential structural change could have severe effects on IBA57 protein stability (which might also result in low protein levels in cells due to protein degradation) or on the interaction with ISCA2, impairing the formation of the heterocomplex and, consequently, the binding of the cluster. In order to further investigate these proposed mechanisms, we plan to conduct in vitro structural and functional characterization of the variant. Specifically, in order to understand if the variant affects the secondary and tertiary structure of IBA57, we are planning to crystallize the variant in its apo form. Additionally, we will investigate the interaction of the Gly104Cys IBA57 variant with ISCA2. If the two proteins interact, forming a heterocomplex, we will test whether a [2Fe-2S] cluster can be chemically reconstituted on it. Furthermore, we will determine if the Cys104 residue promotes aberrant Fe-S cluster coordination by acting as an alternative ligand. Lastly, we will test if the chemically reconstituted Gly104Cys IBA57-[2Fe-2S]-ISCA2 complex is capable of activating apo aconitase in vitro.

Our review of the literature has helped to construct a clearer global picture of a “typical” case of MMDS3 as a previously healthy infant presenting with psychomotor regression and decreased energy with visual impairment, hypotonia, and upper motor neuron signs on exam. Metabolic profiling appears to most consistently reveal lactic acidosis in CSF fluid, whereas serum lactate levels appear to fluctuate during periods of disease exacerbations. MRI most often reveals signal abnormalities in the periventricular white matter with cavitations and lactate peak on MRS. Despite these commonalities, much of the variance in disease severity and progression among affected siblings remains unexplained. Hamanaka et al. (2018) and Vinkler et al. (2003) describe two male siblings affected by compound heterozygous variants. Both presented with acute onset of visual impairment at 18 and 20 months of age, respectively. One sibling presented with stable disease, asymptomatic white matter abnormalities on MRI, and no additional neurologic deficits. His brother, however, developed progressive disease following a Shigella infection and has severe motor and intellectual deficits. Interestingly, despite his severe symptoms, analysis of respiratory chain complex activity did not reveal detectable deficits. Similarly, Zhan et al. (2021) describe two siblings, only one of whom developed progressive disease following vaccination and died shortly thereafter. Interestingly, 56% (14/25) of patients were reported to have a febrile illness or received a vaccine prior to the onset of initial symptoms or prior to an acute exacerbation, which likely reflects metabolic decompensation in the setting of increased catabolic demand. Both of these observations support a significant role of epigenetic and environmental effects on disease course. With respect to the establishment of a genotype-phenotype correlation, apart from the existence of a putative mutational hotspot, we were unable to establish any clear relationship between the variant type or locus with disease severity (data not shown). Interestingly, a splice-site variant in *IBA57* was reported by Lossos et al. (2015), which was shown to cause a SPOAN-like phenotype in 11 members of a consanguineous family (not included in our review). Phenotypically, patients presented with childhood onset of slowly progressive paraplegia, peripheral neuropathy, and optic atrophy. Patients had normal intelligence with 100% survival into adulthood. Neuroimaging did reveal white matter abnormalities in 3/3 patients, including areas of cavitation. Biochemical studies reviewed partially decreased, but not absent, levels of IBA57 protein and complexes I–II of the mitochondrial respiratory chain, as well as a moderate decline of lipoylated pyruvate dehydrogenase and alpha ketoglutarate dehydrogenase.

We feel that our review builds upon a recent comprehensive and well-written review by Lebigot and colleagues (2021) by utilizing a systemic approach to estimate the approximate incidence of clinical and neuroradiographic features across cases and presenting these features in a tabular format easily parsed by a clinician. Additionally, while Lebigot et al. (2021) describe the estimated incidence of some clinical features, this estimate includes data from patients presenting with the SPOAN-like phenotype associated with a splice-site variant, which we believe represents a distinct clinical entity due to its highly attenuated phenotype and later age of onset.

However, our literature review is limited by the non-uniform reporting of patient signs, symptoms, and laboratory data across case reports and case series, as well as limited follow-up periods. In order the better understand the phenotypic spectrum of disease and the natural history of MMDS3, a standardized reporting system is needed. The Newcastle Pediatric Mitochondrial Disease Scale [30] is one such standardized, multi-dimensional, semi-quantitative rating scale designed to facilitate the study of the longitudinal natural history of mitochondrial disease in children and may be of use in future studies.

## Figures and Tables

**Figure 1 genes-13-02044-f001:**
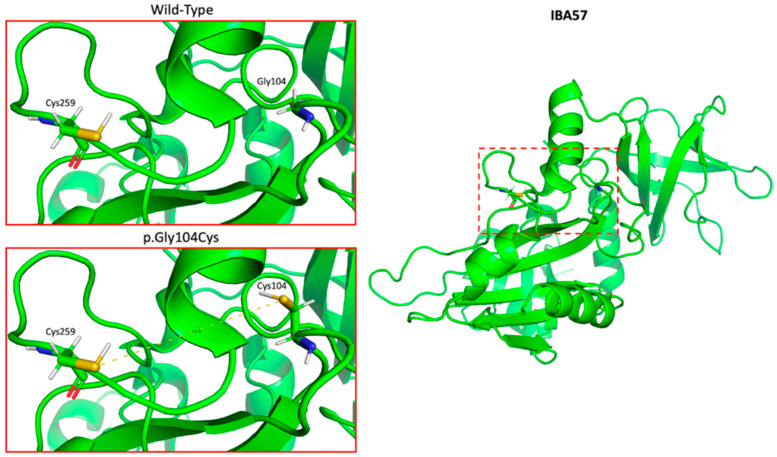
PyMOL model of the crystal structure of IBA57 (RCSB protein databank entry #: 6QE3) showing ~11 Å proximity to the key active Cys259 ligand required for [2Fe-2S] coordination in the context of [2Fe-2S] ISCA2-IBA57 heterocomplex formation.

**Figure 2 genes-13-02044-f002:**
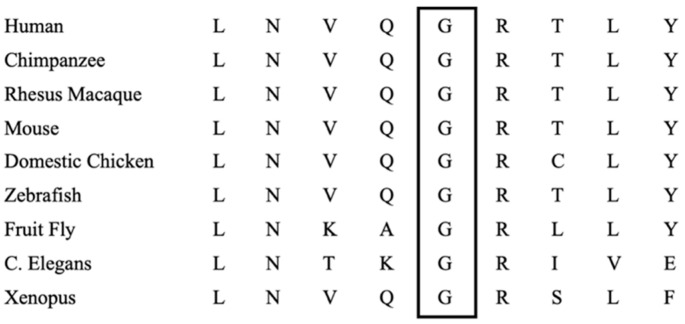
Protein sequence alignment showing 100% conservation of the Gly104 residue (rectangle) across all reported *IBA57* homologs.

**Figure 3 genes-13-02044-f003:**
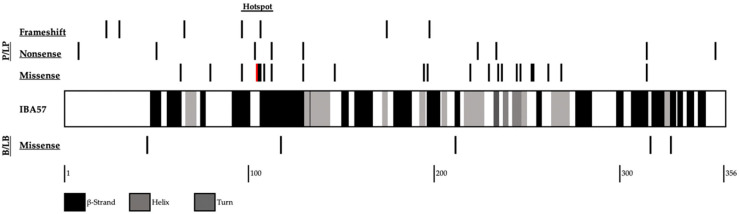
Schematic diagram depicting the mutational landscape of the IBA57 protein. Shaded boxes represent elements of protein secondary structure, and bars depict a previously reported variant at that loci. The p.Gly104Cys variant is shown in red. Pathogenic/likely pathogenic variants (P/LP) were obtained from both existing case reports/series and the ClinVar database. Benign/likely benign (B/LB) variants were obtained from the ClinVar database. The putative mutational hotspot is identified as residues 96–112 (figure adapted from UniprotKB and Lebigot et al. (2021) [6]).

**Figure 4 genes-13-02044-f004:**
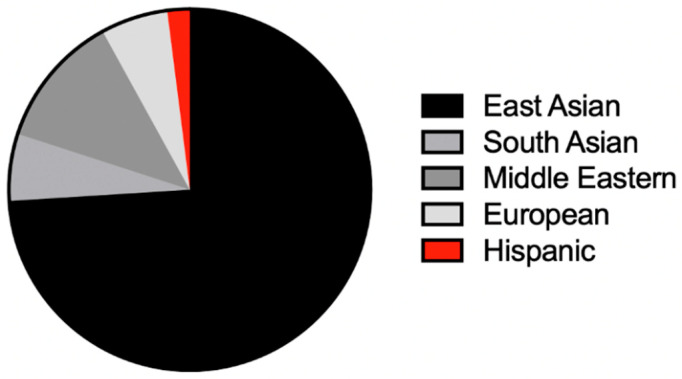
Pie chart depicting proportion of affected patients across major ethnic groups (N = 50). East Asian (Chinese (*n* = 32), Japanese (*n* = 5)), South Asian (Indian (*n* = 3)), Middle Eastern (Moroccan (*n* = 3), Sephardic Jewish (*n* = 2), Tunisian (*n* = 1)), European (Italian (*n* = 3)), and Hispanic (Cuban (*n* = 1)).

**Table 1 genes-13-02044-t001:** ACMG point-based scoring system applied to the (NM_001010867.2): c.310G>T (p.Gly104Cys) variant.

Evidence of Pathogenicity	Category	Evidence	Score
Moderate	PM1	Hotspot of length 17 amino acid residues contains 7 pathogenic variants (2 frameshift, 2 nonsense, and 3 missense), 3 variants of uncertain significance (missense), and no benign variants	2
	PM2	Variant not found in gnomAD genomes	2
	PM3	Detected in trans with a pathogenic variant (Sato et al., 2021)	2
Supporting	PP3	Multiple lines of computational evidence support a deleterious effect on the gene In addition to PyMOL model and MutationTaster used in our manuscript, BayesDel_addAF gives the variant a score of 0.546, which is between 0.421 and 0.625, thus predicting strong pathogenicity	1
	PP4	Patient’s phenotype is highly specific for multiple mitochondrial dysfunctions syndrome-3 (MMDS3), as demonstrated by our review of the literature	1
Total			8

**Table 2 genes-13-02044-t002:** Clinical characteristics of multiple mitochondrial dysfunction syndrome type 3 patients.

Presenting Sign	% (n/N)	Age (Months) [Range]
Psychomotor Regression	81.8 (27/33)	-
Decreased Energy	12.1 (4/33)	-
Hypotonia at Birth	9.1 (3/33)	-
Visual Impairment	6.1 (2/33)	-
Hemiparesis	3 (1/33)	-
**Sex**		
Female	54.9 (28/51)	-
Male	45.1 (23/51)	-
**Neurological Features**		
Impaired Motor Function	97.6 (40/41)	-
Hypotonia	91.7 (11/12)	-
Spasticity	84.6 (11/13)	-
Spastic Tetraparesis	50 (5/10)	-
Intellectual Disability	80 (12/15)	-
Dysphagia	80 (4/5)	-
Muscle Weakness	71.4 (5/7)	-
Limb Contractures	60 (3/5)	-
Nystagmus	60 (3/5)	-
Epilepsy	35 (7/20)	-
**Ophthalmologic Features**		
Visual Impairment	46.7 (7/15)	-
**Biochemical Findings**		
Elevated CSF Lactate	72.7 (8/11)	-
Elevated Serum Glycine	60 (6/10)	-
Elevated Serum Lactate	53.8 (7/13)	-
**Other**		-
Feeding Difficulty	100 (6/6)	-
Apneic Episodes/Respiratory Failure	100 (6/6)	-
**Natural History**		
Normal Development Prior to Presentation	67.9 (19/28)	-
Median Age Of Symptom Onset, N = 52	-	9 (0–22)
Median Age at Last Evaluation, N = 30	-	27 (0–348)
Median Age of Death, N = 10	-	7 (0–27)
Preceding Infection/Vaccine	56 (14/25)	-
Stable/Improvement	27.3 (6/22)	-
Paroxysmal Worsening	22.7 (5/22)	-
Progressive Worsening	50 (11/22)	-

**Table 3 genes-13-02044-t003:** Neuroradiographic characteristics of multiple mitochondrial dysfunction syndrome type 3 patients.

MRI Findings	
White Matter Lesion Location	% (n/N)
Periventricular	83.3 (20/24)
Parieto-occipital White Matter	66.7 (14/21)
Corpus Callosum	62.5 (15/24)
Splenium	38.5 (5/13)
Spinal Cord	41.7 (5/12)
Frontal White Matter	30 (3/10)
Brainstem	28.6 (4/14)
Cerebellum	25.9 (7/27)
Internal Capsule	22.7 (5/22)
Temporal White Matter	19 (4/21)
Thalamus	10 (1/10)
Optic Nerves	10 (1/10)
**Other**	
Lactate Peak on MRS	100 (9/9)
Presence of Cavitations	81.5 (22/27)

## Data Availability

The data presented in this study are available in publicly accessible repositories. The protein structure of IBA57 presented in this study is openly available in the Protein Data Bank at https://www.rcsb.org/structure/6QE3 accessed on 1 August 2022.

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
