# Peer review of "Multiple Mitochondrial Dysfunction Syndrome Type 3: A Likely Pathogenic Homozygous Variant Affecting a Patient of Cuban Descent and Literature Review"

_genes, 2022, doi:10.3390/genes13112044_

Round 1

Reviewer 1 Report

Please find the comments regarding the revision of the manuscript entitled Multiple Mitochondrial Dysfunction Syndrome Type 3: A novel homozygous variant and literature review”.

In this manuscript Lang and co-workers report a patient with Multiple Mitochondrial Dysfunction Syndrome Type 3 (MMDS3) carrying a novel homozygous mutation in IBA57. In addition, the authors’ provide a comprehensive bibliographic review compiling the main clinical and biochemical characteristics of MMDS3 patients. The clinical data is well presented and the review is interesting, as it provides a global picture of this disorder.

However, in my opinion, the present manuscript is, in fact, a mixture of two different studies: (i) a description of a new patient with a novel IBA57 mutation, and (ii) a literature review of MMDS3.

(i) The description of a new patient with a novel mutation. Although “in silico” protein structure analysis is provided, a functional test is still lacking to unequivocally demonstrate that the IBA57 variants are the cause of the disease in this individual, e.g, western blot of IBA57 to determine protein levels, or analysis of lipoic acid. If patient material is not available to perform such studies as an alternative I would strongly suggest a comprehensive “in silico” analysis using already reported IBA57 mutations in which functional data is available, in order to correlate “in silico” defects with real laboratory studies. This point could give some novelty to the manuscript.

(ii) The review of MMDS3 is interesting and the data is well presented. However, a recent review of Fe-S defects (including those associated to IBA57 deficiency) has been published (Lebigot, E., Schiff, M., & Golinelli-Cohen, M. P. (2021). A Review of Multiple Mitochondrial Dysfunction Syndromes, Syndromes Associated with Defective Fe-S Protein Maturation. Biomedicines, 9(8), 989). The present revision should be contextualized with the previous one.

Reviewer 2 Report

General Remark :

The manuscript describes a patient carrying a novel homozygous variant in IBA57 . The originality of this report is limited but it would contribute to expand the genotype and phenotype of patients with MMDS type 3. The major limit of the manuscript is the lack of arguments to confirm the pathogenicity of the variant identified in IBA57 gene.

Major comments :

The authors classified the homozygous variant identified in IBA57 as pathogenic. However, clinical interpretation of genetic variants by ACMG classifies the variant as variant of unknown significance (VUS) with criteria PM2 and PM3. Any biochemical data may help to upgrade the variant as likely pathogenic.

Structural modeling suggests the loss of the interaction between the modified amino acid and a residue, essential for IBA57 function. This interpretation is hypothetical and remains insufficient to confirm deleterious effect of the identified variant.

The authors do not notice if tissues of the patient are available to perform further analysis and also confirm the causative effect of IBA57 variant on mitochondrial proteins dysfunction (respiratory chain complexes, PDH or KGDH activities, levels of mitochondrial lipoylated protein and of BA57 protein…). If tissues of the patient are not available, the authors have to perform other functional studies such as complementation assays for example.

Minor Comments:

11)      Why do the authors not include patients with SPOAN-like phenotype in the review ? It would be appreciate to review all patients with MMDS3.

22)      In the materials and methods section, authors write that WES trio analysis was performed. However, variants of interest are confirmed by Sanger analysis in proband’s parents. Was the exome analysis only performed for the proband or in trio ? The explanation must be reformulated.

3)      3) Concerning clinical data, it would be interesting to add the weight, height and head circumference at birth of the patient.

The authors write several times « previously healthy 2 months-old girl », then they explain that hypotonia began one month earlier. So, this expression is false and it would be better to delete « previously healthy 2 months-old girl » in the manuscript.

44)      If available, it would be interesting to note the blood lactate level during lactic acidosis.

55)      Table 1 and table 2: it would be clearer to position the data « n » et « N » near the percentage. That is, for the 1st line « 81.8% (27/33) ». Furthermore, the results in months (median age of XX), are hard to read, so a dedicated column would be more lisible for these data.

66)      Table 2, last line : « presence » instead of « prescence »

Round 2

Reviewer 1 Report

The main issues required have been properly addressed. Although functional laboratory data is finally not provided (due to limited patient' tissue availability) in silico studies support the pathogenic effect of the mutation. 

I have no more to comments or suggestions.

Reviewer 2 Report

Lines 137 , 138, 143 : percentile in letters

Lines 151-152 : please add reference value of blood lactate level in the lab

Lines 167 and 310: ansgtröm instead of angstrom

Line 177 : there is a missing bracket

Line 183 : the variant R105W is downstream of your variant . 

Lines 183 - 192 : this paragraph must be summarized.

Line 193 : "With respect to the.."

Table 1 : "PP3" ; the word "manuscript" is misspelled

Table 3: last line : « presence » instead of « prescence »

Line 351 :  "Shigella" instead of "shigella"

Line 369 : delete italics for "IBA57" . Add complexes I and II "of mitochondrial respiratory chain"

Line 378 : the sentence "Thus, inclusion of the SPOAN like patients would confound analysis of the more severe “typical” MMDS3 phenotype caused by the 379 biallelic missense, nonsense, and frameshift variants affecting the infants and toddlers included in our review" could be deleted
